# Physical, Chemical and Biochemical Modification Approaches of Potato (Peel) Constituents for Bio-Based Food Packaging Concepts: A Review

**DOI:** 10.3390/foods11182927

**Published:** 2022-09-19

**Authors:** Katharina Miller, Corina L. Reichert, Markus Schmid, Myriam Loeffler

**Affiliations:** 1Research Group: Meat Technology & Science of Protein-Rich Foods (MTSP), Department of Microbial and Molecular Systems, Leuven Food Science and Nutrition Research Centre, KU Leuven Ghent Technology Campus, B-9000 Ghent, Belgium or; 2Sustainable Packaging Institute SPI, Faculty of Life Sciences, Albstadt-Sigmaringen University, 72488 Sigmaringen, Germany

**Keywords:** potato starch, potato protein, potato peel-based films, biopolymer modification

## Abstract

Potatoes are grown in large quantities and are mainly used as food or animal feed. Potato processing generates a large amount of side streams, which are currently low value by-products of the potato processing industry. The utilization of the potato peel side stream and other potato residues is also becoming increasingly important from a sustainability point of view. Individual constituents of potato peel or complete potato tubers can for instance be used for application in other products such as bio-based food packaging. Prior using constituents for specific applications, their properties and characteristics need to be known and understood. This article extensively reviews the scientific literature about physical, chemical, and biochemical modification of potato constituents. Besides short explanations about the modification techniques, extensive summaries of the results from scientific articles are outlined focusing on the main constituents of potatoes, namely potato starch and potato protein. The effects of the different modification techniques are qualitatively interpreted in tables to obtain a condensed overview about the influence of different modification techniques on the potato constituents. Overall, this article provides an up-to-date and comprehensive overview of the possibilities and implications of modifying potato components for potential further valorization in, e.g., bio-based food packaging.

## 1. Introduction

Potatoes are one of the four major food crops worldwide (first rice, second wheat, third corn, fourth potato) [1]. In the European Union (EU-28), the total potato harvest was 56.4 million tons in 2019. Especially in Europe, potato cultivation and the potato industry are very important in the domestic food culture and diet, and the increasing consumption of processed potatoes in the form of potato chips, French fries, mashed potatoes, etc., represents a worldwide trend [2]. In comparison, the annual potato harvest in China in 2019 was 91.8 million tons. China is thus the largest potato-producing country in the world [3].

Within the potato industry, approximately half of the harvested potatoes become side streams or residual material. Thus, the entire supply chain (including farmers, wholesalers, the processing industry, retailers and consumers) produces tubers, peels and pulp that are discarded [4], as illustrated in Figure 1 for non-organic processing potatoes. 

To summarize the findings, the product losses along the supply chain of non-organic processing potatoes in Switzerland accounts for about 44%, which means that from an initial input of approximately 1.8 kg potatoes one receives an output of 1 kg processed potato product such as French fries [5]. Along the supply chain of processing potatoes, main losses can be attributed to quality standards (21.9%), followed by the potato peel side stream (10.1%) [4]. Similar data on the amount of side streams and losses along the potato supply chain can be found in Germany [6] and Europe in general [7]. Quality standards lead to a high amount of potato-based side streams or residuals, which in part are determined by the potato processing industry. For instance, PepsiCo has specific processing quality parameters in their potato crisp processing manufacturing in India to control, amongst others, tuber size, dry matter content, sugar content, starch content, damage and discoloration [8]. Besides the amount of potatoes that are lost based on quality aspects, there is also a high amount of potato peel. In Figure 1, the loss attributed to peeling is 10.1% based on the initial crop yield. If attributed to the input quantity at the processing stage (69.8% of initial crop yield) losses attributed to peeling would amount to 15.5%. In the literature, potato peel side stream losses of up to 40% have been reported which varied depending on the peeling method used [9]. The large amount of potato side streams led to the establishment of different strategies to valorize them [10].

Microbial spoilage of potato side stream products and widespread production locations result in low utilization of these byproducts. In Switzerland, most of the potato side streams (90%) are used as animal feed due to the high protein content [4] corroborating with data from other countries [1]. In accordance with the principles of a circular bio-economy, the use of potato side streams is of great interest for the production of high quality products with zero-waste generation [11,12]. As these large amounts of side streams currently possess low or even negative economic value potential, valorization approaches of potato side streams are becoming important [13]. 

One valorization approach for potato side streams is the production of bioplastics [14]. Due to environmental challenges caused by large amounts of petrochemical-based and not biodegradable packaging, the utilization of bio-based alternatives is demanded. For instance, potato side streams such as potato peel or potato pulp can be used for potato starch or potato protein extraction which can be further processed to bio-based plastics [15,16,17,18,19,20,21]. However, extraction processes are energy and time consuming and result in additional side stream productions. A material efficient approach represents the use of the whole side stream without extracting single constituents. Several studies demonstrated that mechanical and barrier properties of potato flour- or potato peel-based films are not inferior to the properties of films composed of potato starch or potato protein suggesting that extensive industrial purification may not be necessary for film production [15,17,18,22,23,24]. 

To successfully implement a circular bio-economy concept, the quality and characteristics of alternative materials (in this case biopolymers made from potato side streams) must be at least as good as their commercial equivalent (in this case petrochemical plastics) [11]. At the moment, potato side stream-based biopolymers are characterized by an overall low technology readiness level and in part insufficient processing- and packaging related properties including processability and mechanical properties [23,25]. One approach to increase the technology readiness level and optimize processing- and packaging related properties of potato side stream-based films and coatings can be physical, chemical or biochemical modification which are outlined and discussed in this review article. 

Based on previously conducted studies, some modification methods have been used to influence physicochemical properties of potato peel-based films including high-pressure homogenization, ultrasonication, gamma irradiation and acid hydrolysis. This review aims to summarize and discuss the influence of different physical, chemical and biochemical modification methods on the structure and physicochemical properties of potato constituents (mainly potato starch and potato protein) and their films. In this review, different modification methods of potato constituents are evaluated upon their suitability to change a certain property especially in terms of the modification effect on the film-forming and packaging related properties such as rheological, thermal, and barrier characteristics. Besides a comprehensive review, we highlight recent research gaps and future research potentials of potato constituents for packaging applications.

## 2. Potato Tubers

Potato tubers are the stems of the potato plant that develop under the soil from the propagation of previously planted mother tubers by sprouting. Sprouting occurs under suitable conditions from the dormant/lateral buds of the potato tubers (also known as “potato eyes”). Although shape, color and size vary between different cultivars, the morphology of potato tubers is generally characterized by an oval to round shape, white flesh and light brown skin. The potato flesh consists of the pith, perimedullary zone, vascular ring and cortex (Figure 2), which are usually consumed by humans. The periderm (skin) contains the apical bud, stem end, lenticels and lateral buds, as shown in Figure 2. In addition to the type of potato cultivars, composition of potato tubers also depends on several other factors including soil type and temperature, location, cultural practices, maturity and postharvest storage conditions. On average, potato tubers mostly consist of water (approximately 80%) and 20% dry matter. Related to the dry matter content, carbohydrates, especially starch (see Section 2.1), are the main fraction, followed by proteins (see Section 2.2), which can be both extracted from potato tubers or their side streams [26,27,28].

Based on its high dry-matter starch content (Section 2.1), potato tubers are favorably used by the industry for starch extraction. In laboratory scale experiments, starch extraction is often conducted via sedimentation or centrifugation method [15,29,30]. Therefore, potato tubers are usually peeled and washed to remove defective parts and contaminations, and then grounded to destroy cell structures and release starch granules. The obtained slurry is filtered and washed to further remove contaminants. Based on the higher density and low cold-water solubility most of the starch can be removed as a sediment using decantation of the supernatant prior drying to obtain starch powder. Alternatively, the separation between starch and supernatant can be performed by centrifugation. Potato starch can also be extracted from potato side streams including potato peels [15,29,30].

Accordingly, potato protein (Section 2.2) can be extracted either from potato tubers or potato side streams. For instance, after potato starch and tuber cell (pulp) removal, the supernatant (potato juice) of the centrifuged sample, can be further used for protein recovery [31]. To increase the concentration of potato protein, purification by ultrafiltration, reverse osmosis or fractional precipitation can be conducted. Precipitation methods generally include salting out, heat denaturation and isoelectric precipitation [32,33,34]. In laboratory scale experiments, potato protein is recovered at pH 3.0 or 5.5 under thermal treatment [35]. However, dependent on the extraction and purification methods, the recovered yield of potato protein and its composition (e.g., amount of patatin, protease inhibitors and high molecular weight proteins) vary [36].

### 2.1. Potato Starch

With 70 to 80% *w*/*w* of dry matter content, starch is the major component of potato tubers. The structure of starch on molecular to macromolecular level is schematically illustrated in its main levels in Figure 3. The structure of potato starch typically consists of 20 to 30% (*w*/*w*) amylose, which is a linear to slightly branched (degree of polymerization > 60) polymer chain consisting of α-d-(1,4)-linked glucose units (Figure 3a), and 70 to 80% (*w*/*w*) of amylopectin, which is a highly branched polymer based on relatively short chains of α-d-(1,4)-linked glucose units with 4 to 6% (*w*/*w*) additional α-d-(1,6)-linkages (Figure 3b). Low molecular weight amylose (0.13–0.5 MDa) is assumed to form a helical structure (Figure 3c). Amylopectin is characterized by its high molecular weight (10–1000 MDa) and its branched structure, with branched chains being classified by their substitution into A chains (not substituted with other chains), and B chains (substituted with other chains) (Figure 3d). These branched chains derive from the main chain, which is called C chain (Figure 3d). The reducing end of the amylopectin polymer can be found at the end of the C chain (Figure 3d). Due to the tight packing of A and B chains, they can form double helices with six glucose units building one turn (Figure 3e). The arrangement of interlinked clusters results in lamellar structures, where alternating amorphous (branching points/interconnecting chains) and (semi-) crystalline (A and B chain clusters) parts are present (Figure 3f). On a macromolecular level, several of the alternating lamellar structures are organized in so called growth rings forming the starch granules (Figure 3g). The amorphous center of the granule (hilum) is formed mainly by amylose chains (Figure 3h) [37,38,39].

Potato starch granules have a spherical to oval shape with a diameter of 10 to 100 µm and a smooth surface. Furthermore, potato starch is usually characterized by its B type arrangement of double helices, which can be observed in X-ray diffraction analysis based on peaks at 5.6°, 17°, 22° and 24° (2θ) and a relative crystallinity of approx. 30%. In contrast to A type starch (peaks at 15°, 17°, 18° and 23° (2θ)), B type polymorphs basically have a more open structure and contain more water molecules [40]. In addition to A and B type polymorphs, there are C type polymorphs, which represent a mixture of the A and B type. V-type structures are composed of complexes between amylose single, left-handed helices and complexing compounds, which fill the helical cavities of the amylose in the presence of water [41,42]. One example are amylose–lipid complexes, which can be found in native starch. In those, lipid chains fill in the central cavity of amylose helices by interacting with the hydrophobic moiety of the amylose chain. These complexes can affect different properties of the starch and their films [43]. V-type crystallinity can also occur during starch retrogradation [44]. Another characteristic of starches is their phosphorous content, which was found to correlate in reverse with the amylose content and influences pasting and retrogradation properties. In comparison to other starches, potato starch contains high amounts of phosphorous referring to a high amylopectin content and long amylopectin chains. Mostly, the phosphorous is covalently bound to the amylopectin fraction (ester of phosphoric acid and hydroxyl group) of potato starch [45,46]. Regarding functional properties, the onset gelatinization temperature of potato starch is around 58–63 °C, dependent on its cultivar and crystalline order [47,48]. Amongst various starch sources, potato starches are known for its high swelling power and pasting viscosity, enabling the formation of thick viscoelastic gels [49].

### 2.2. Potato Protein

Although potato proteins account for only 1 to 3% of the fresh weight of potato tubers, they are the second largest fraction of the dry matter of potato tubers. Potato proteins consist of water insoluble and water-soluble proteins, which account for 25 and 75% of potato tuber proteins, respectively. Based on their function, soluble potato proteins are classified into three groups, namely patatins (30–40%), protease inhibitors (40–50%) and other high molecular weight proteins (10–15%) [39,50].

Patatins (Figure 4a) are storage proteins, consisting of homologous glycoprotein isoforms. They are characterized by molecular weights of 40–45 kDa and vary in their isoelectric points between 4.5–5.2. The secondary structure of patatin consists of approximately 33% α-helices, 45% β-sheets and 15% random turns [51]. Stability of patatin decreases with temperatures above 45 °C and the denaturation temperature was measured to be between 59–60 °C. The best gelatinization of patatin was reported to be at pH values between 4.8–5.5 [52]. At a pH of 4, the solubility of patatin is minimal, and at a pH below 4, irreversible unfolding of the tertiary and secondary structure of patatin can occur. Patatins are known for their emulsifying, foaming, gel forming or antioxidant properties, making them attractive for food and non-food applications [53,54,55].

Protease inhibitors (Figure 4b) are a heterogeneous group with varying reactive groups. Protease inhibitors are classified into four major sub-groups, namely serine-, cysteine-, metalloid- and aspartic protease inhibitors. Protease inhibitors have molecular weights of approximately 5–25 kDa and differ in their isoelectric points between 5.1–9.0 [56]. The gelatinization optimum for protease inhibitors was reported to be at pH 3.2–4.3 [52]. With approximately 24%, serine protease inhibitors are the biggest fraction within the group of protease inhibitors in potato proteins. The denaturation temperature of serine protease inhibitors is about 69 °C, and their secondary structure is composed of about 2% α-helices, 38% β-sheets, 23% β-turns and 37% random coils [57]. Compared to patatin, protease inhibitors are more hydrophilic and were reported to have superior antifungal and antimicrobial properties [36,39,50,56].

Other high molecular weight proteins (>40 kDa) are the third group of proteins present in potato tubers. Based on their relative low percentage as mentioned above, the group of other high molecular weight potato proteins is not as well described as patatin and protease inhibitors. However, it has been reported that this group contains among others polyphenol oxidases, lipoxygenases and starch synthesis associated enzymes [39,50,55]. 

In addition to the mentioned potato protein fractions, potatoes also contain free amino acids, which account for up to 40% of the total nitrogen content. Amino acid composition of potato protein does not only play a crucial role in dietary/nutrition [39] but also determines modification potentials, as the composition impacts available functional groups of the amino acid side chains [60]. Table 1 shows the amino acid composition of potato proteins with their associated functional groups, ordered from the smallest to the biggest fraction.

## 3. Modifications

Depending on the nature of changes induced, modification of molecules or polymers can be divided into physical (Section 3.1), chemical (Section 3.2) or biochemical (Section 3.3) modifications, which are subsequently described for potato constituents. Besides individual modification techniques, combinations of modifications can be applied. These modifications are often referred to as dual modifications, and are also described below (Section 3.4).

### 3.1. Physical Modification

One approach to alter the physicochemical properties of powders including starches and proteins is via physical treatment (e.g., hydrothermal treatment, irradiation, ultrasonication and high-pressure treatment, see Section 3.1.1 and Section 3.1.2). Besides chemical modification, starches and proteins, which have been modified via physical methods do not have to be claimed as “modified”. Physical modification is also viewed as cost-efficient and environmentally friendly, because no hazardous substances (chemicals) are used. Among the various physical modifications, a distinction is made between thermal (Section 3.1.1) and non-thermal treatment methods (Section 3.1.2) [62].

The physical modification of starch has already been reviewed by several authors [62,63,64,65,66]. However, due to fundamental differences in structure, morphology and physicochemical properties of physically modified starches from different botanical sources (e.g., corn, maize, tapioca, rice, pea, potato), it is important to specifically review the literature on physical modifications of potato starch [67,68,69,70]. So far, the physical modification of potato starch has been reviewed elsewhere describing the effect from emerging technologies, including high pressure, ultrasound and microwaves, cold plasma and electric pulse [71]. 

#### 3.1.1. Thermal Physical Modification

In this review article, thermal physical modification of a substance is defined as a physical treatment, which generates or transfers heat or cold onto a substance in the presence or absence of moisture to create structural and/or functional changes. Therefore, thermal methods to modify potato starch include hydrothermal methods such as heat moisture treatment and annealing, dielectric treatments such as microwave heating and radio frequency heating, as well as other methods including freeze-thaw treatment, dry heat treatment, flash drying, autoclaving and modification via superheated steam.

##### Heat Moisture Treatment

Heat moisture treatment (HMT) refers to the treatment of a product with heat in combination with a certain amount of moisture. Thus, the parameters for temperature, time, moisture and cycles/repetitions are crucial for this treatment method.

In HMT the amount of water in the treated samples is limited to 10–35%, to prevent starch gelatinization [72]. Therefore, granular morphology of starch is preserved during HMT [73]. However, several authors noted that HMT leads to some granular damage [67,73]. The temperature applied during HMT is characterized being above the gelatinization temperature, but below the glass transition temperature. For HMT of potato starch temperatures of 80–121 °C, durations of 30 min to 16 h, and iterations of up to six have been applied in the last years [29,74,75].

Overall, the thermal properties, such as gelatinization temperatures (onset (To), peak (Tp) and conclusion (Tc)) of potato starch significantly increased and melting enthalpy (ΔH) decreased by HMT. This increase in thermal properties was accelerated and/or intensified with increasing temperature [67], increasing holding time [74] and decreasing moisture content [75]. Shi et al. [67] linked the changes in gelatinization temperatures with increasing HMT temperatures to intramolecular interactions towards more amylose–amylose and amylose–lipid interactions at higher temperatures, which hindered the mobility of the amorphous region thus increasing the gelatinization temperature. Decreased mobility leads to decreased water uptake and swelling, which is necessary to facilitate melting of crystals and double helices. Therefore, stability of the crystalline regions of potato starch increased and higher temperatures for gelatinization were required after HMT. Accordingly, amylose content increased, degree of crystallinity decreased and type of crystallinity varied upon HMT of potato starch, using different temperatures and moisture contents [67,73,75,76]. Based on the reviewed studies it seems that the change in type of crystallinity of potato starch from B to C or A, which occurs at 120 °C, depends on the moisture content of the material: C-type occurred at 30% [73], but A-type at 10–27% [29,67,75] moisture content. At lower temperature (100 °C) a change from B to A-type crystallinity occurred at 24% moisture content [76]. 

Hydration and pasting properties of potato starch were also affected by HMT. Cold water solubility of potato starch increased whereas swelling power, water absorption capacity and hot water solubility decreased [20,29,73,74,76]. Upon HMT, pasting temperature increased with increasing temperature [29]. Through HMT, the final viscosity decreased with increasing temperature [29] and increasing holding time [74], but final viscosity increased at short holding times [74] and temperatures < 120 °C [29]. 

Concerning potato starch as packaging material, HMT did not significantly affect packaging related properties of potato starch-based bioplastic [15].

##### Annealing

Annealing (ANN) is another method to physically modify starch, without destroying its granular morphology. In contrast to HMT, moisture content of the samples is not limited (usually being > 60%, *w*/*w*) and temperature used during ANN is below the sample To (onset gelatinization temperature). Common processing conditions during ANN of potato starch are 30 to 55 °C for 12 h to 96 h. Here, iterations of ANN treatment cycles can be applied [20,77,78]. 

Overall, ANN treatment of potato starch increased gelatinization temperatures (To, Tp, Tc), had no effect on type of crystallinity and granule surface, but increased pasting properties (pasting temperature and final viscosity) [20,77,78] (see Table 2), which were favored by higher temperatures [77] and longer treatment times/iterations [78] used for ANN treatment. Furthermore, increased hot water solubility and hot water swelling power of ANN treated potato starch granules were reported [78]. Through processing of starch in water having temperatures between 50 and 70 °C, solubility and swelling power of ANN potato starch was lower than for native potato starch [78]. Wang et al. [77] and Xu et al. [78] reported an increase in relative crystallinity, increase in ΔH, as well as changes in absorbance of potato starch, which might be attributed to the higher ANN temperature, treatment times or suspension concentration used (55 °C vs. 30–50 °C, 12–96 h vs. 24 h, 1:3 *w*/*v* vs. 1:5 *w*/*v*). 

##### Microwave Treatment

Microwave (MW) treatment is a thermal heating method using microwaves (300 MHz to 300 GHz), that can be absorbed by polar materials, such as potato starch. Upon MW heating, molecular vibrations and frictions occur due to their alignment to the vibrating electromagnetic field, resulting in the generation of heat within the product. Therefore, MW heating requires less treatment time and can be more efficient and less energy consuming than conventional heating [79,80]. The ability of a product to be heated via MW treatment is dependent on moisture content, frequency, as well as compaction density and temperature of the sample [81,82,83]. The influence of MW treatment on starch structure and starch properties has been recently reviewed [84,85]. In addition, Jiang et al. [86] reviewed the effects of MW treatment on different food components, including proteins. In contrast to other thermal modification methods such as HMT or ANN, MW treatment is not limited to certain moisture or temperature and can be therefore performed not only for potato starch powders [68,87,88], but also in potato tubers [89] or for starch–water suspensions [30,90].

Miernik and Jakubowski [89] found that MW treatment of potato tubers show a strong correlation between MW dose and starch granule size, which resembled a parabola with its minimal turning point at 53.19 J/g. This means that granule size increased, when MW doses of 0–47.32 J/g or 66.14–508.47 J/g were applied.

Compared to native potato starch granules, MW treated potato starch granules showed damaged structure or severe cracking, which correlated to an increased water absorption capacity [88]. For potato starch with low moisture content (16.5%) decreasing gelatinization temperatures and decreasing final viscosities have been reported after MW treatment [87]. Whereas increasing To, decreasing relative crystallinity and increasing pasting properties (pasting temperature and final viscosity) have been observed for MW treated potato starch with higher moisture contents (21% and 30%) [68,88]. Furthermore, changes in relative crystallinity, gelatinization temperatures (To, Tp, Tc) and pasting properties seem to be accelerated with increasing MW intensity and treatment time [68,87,88]. Xu et al. [68] reported a change in crystallinity from B to A-type at high power (6.63 W/g) MW treatment. 

In potato starch solutions, decreased amylose content, increased relative crystallinity, increased gelatinization properties, increased pasting properties, decreased water solubility and increased swelling power were observed after MW treatment [30]. However, Xia et al. [90] did not observe any changes in gelatinization and pasting temperatures in potato starch solutions, but changes in hydration and pasting properties were also observed in potato starch solutions containing added protein [91]. 

Based on the reviewed studies, once can conclude that the effect of MW treatment on potato constituent structure and properties can vary, depending on the processing conditions (moisture content, power, treatment time, cool-off period, etc.) applied [30,68,87,88,89,90,91]. 

##### Radio Frequency Treatment

Similar to MW heating, radio frequency (RF) are electromagnetic waves in the frequency range of 1 to 300 MHz. In comparison to MW, RF has been reported to penetrate products even more deeply and distribute heat more uniformly.

In addition to MW treatment, which is reviewed above, Xia et al. [90] also treated potato starch–water suspensions with RF, varying the water content. Upon RF treatment small fractures on starch granule surfaces were observed, which were less severe than for MW treated starch granules. Similar to MW treated samples, RF treated samples showed lower amylose contents, but relative crystallinity decreased after RF treatment and with increasing water content. In contrast to MW treatment, type of crystallinity of potato starch samples changed from B-type to C-type after RF treatment. Compared to native potato starch, gelatinization temperatures, pasting temperature and ΔH slightly decreased and final viscosity slightly increased after RF treatment [90]. An increase in pasting properties was also reported by Zhu et al. [92], who investigated the influence of RF drying on potato flour properties. However, no changes in sample color, relative crystallinity, thermal properties or type of crystallinity were observed in this study.

##### Others

In addition to hydrothermal (HMT, ANN) and dielectric (MW, RF) treatment methods, we found other thermal modification methods including freezing and thawing (FT), flash drying, autoclaving or steam heating, which can be used to alter potato starch, flour and protein properties. In general, there are methods that can be used in the presence or absence of water to generate/transfer heat.

FT can be applied to induce structural and functional changes in potato constituents. For instance, Zhang et al. [93] investigated the effect of FT treatment on potato starch showing disruption and aggregation of granule fragments after FT treatment. Furthermore, pasting temperature slightly increased and final viscosity of potato starch slightly decreased. A decrease in gelatinization properties (To, Tp and ΔH) after FT treatment was reported by Wang et al. [94], indicating a weakening of the crystalline structure. Another study demonstrated the influence of the thawing method on structural and functional properties of pre-gelatinized potato flour samples, showing an overall increase in gelatinization temperatures and pasting properties upon FT treatment [80].

Similarly, an increase in gelatinization properties (To, Tp, Tc), final viscosity, and a reduction in granule damage and gelatinization were achieved in potato flour upon flash drying (130–135 °C, 1–2 min) as compared to commercial potato flakes and granules [95]. 

The influence of superheated steam (100–160 °C) on potato starch properties was investigated by Hu et al. [96], showing minor effects on granule surface, compared to native starch granules. The type of crystalline structure was not affected by the superheated steam treatment, but relative crystallinity decreased progressively with increasing temperature. Furthermore, thermal properties (To, Tp, Tc and ΔH), solubility and swelling power decreased, and pasting properties increased upon superheated steam treatment [96].

Drozłowska et al. [97] gelatinized potato protein–water solutions (10% *w*/*w*) at 100 °C for 15 or 30 min prior autoclaving the samples at 126 °C for 60 or 120 min to induce thermal hydrolysis. No influence on solubility was observed for potato protein samples after thermal hydrolysis treatment. Viscosity of modified samples significantly increased with decreasing pre-treatment time and increasing autoclaving time compared to native and only pre-treated samples. Finally, emulsion properties of potato protein were slightly increased due to modification.

#### 3.1.2. Non-Thermal Physical Modification

In this subsection, non-thermal physical modification methods are reviewed. Non-thermal physical modification refers to physical treatments that affect structural and/or functional properties of materials without the intentional use of heat or cold. In non-thermal treatment, heat is sometimes generated as a “side stream” (e.g., during high pressure treatment or ultrasonication due to friction, etc.). For instance, high pressure, ultrasound, ionizing irradiation, milling and electric field treatment are non-thermal techniques that are outlined in the following for potato constituents.

##### High Pressure Treatment

High pressure (HP) treatment (30–1000 MPa) is a non-thermal modification method, which can cause starch gelatinization and protein denaturation/gelation [98,99]. Thereby, HP can be applied in different ways, which are commonly referred to as high hydrostatic pressure/high isostatic pressure, high pressure homogenization and dynamic high pressure microfluidization. During high hydrostatic pressure/high isostatic pressure modification of starch and protein, pressure is transmitted uniformly through a pressure-transmitting medium (e.g., water) to the sample (sample-water suspensions), which is usually packed in vacuum bags and placed inside the high pressure vessel [69,100]. The high hydrostatic pressure treatment of starch in its granular form has been previously reported [101]. During high pressure homogenization a liquid sample is pressed through a small orifice causing shear forces. In dynamic high pressure microfluidization the pressure is transferred in an interaction chamber [102,103]. Similar to other modification methods, the effect of HP treatment can vary for starch from different plant sources as reviewed elsewhere [104,105].

Different effects and partially controversial findings of HP treatment on potato starch properties (thermal properties, pasting properties, crystallinity, granular/surface damaging) and potato protein properties (solubility, foam stability) were reported in the reviewed literature [69,70,100,101,106,107,108], which might be attributed to the different types of pressure (high hydrostatic pressure, high isostatic pressure, microfluidization) used. For instance, granular structure of potato starch as well as granule surface were only slightly affected by high isostatic/hydrostatic pressure application [69,70], but significantly damaged by HP microfluidization [106]. Furthermore, within similar HP application and starch–water solution ranges, different effects of HP treatment on potato starch crystallinity, gelatinization properties and pasting properties have been reported in different studies [69,70,101]. Wang et al. [69] did not observe any significant changes in gelatinization temperatures and ΔH, whereas Rahman et al. [70] found increasing gelatinization temperatures with increasing pressure applied to potato starch–water suspensions (1:5 *w*/*w*) with HP (100–500 MPa vs. 200–600 MPa). In contrast, Słomińska et al. [101] observed decreasing gelatinization temperatures for potato starch granules treated at 50–1000 MPa HP, which was accelerated with increasing pressure applied. Rahman et al. [70] additionally investigated hydration properties. A decrease in solubility and swelling power of potato starch was observed upon HP treatment.

For potato proteins, differences in purity and pH did likely contribute to observed qualitative and quantitative differences in some functional properties throughout the reviewed studies. For instance, it seems that physical changes by HP treatment cannot be induced at pH 7 but likely at pH < 7 [100,108]. This was shown by Baier and Knorr [100] who observed a significant decrease in solubility and an increase in surface hydrophobicity for patatin fractions that were HP treated at pH 6, but no effect were observed at pH 7. In addition, Katzav et al. [108] reported that potato protein gelation was only possible when samples were HP treated at pH 3 but not at pH 7, independent of the amount of pressure (300–500 MPa) applied.

Upon HP treatment at pH 3.6, the secondary structure of potato protein was significantly affected as an increase in α-helix content and decrease in β-sheet structures has been determined by Hu et al. [107]. Furthermore, particle size and emulsion stability of potato protein stabilized emulsions decreased and foaming properties slightly increased. In contrast to the patatin fraction studied by Baier and Knorr [100], Hu et al. [107] reported increased solubility of potato protein upon HP treatment.

Kang et al. [25] investigated the effect of HP treatment on the film-forming properties of potato peel which are used for film production. Compared to native film-forming potato peel solutions, HP treated solutions showed higher elastic modulus and viscosity at higher temperatures (>60 °C). Increasing viscosity values with increasing pressure and number of passes were observed for HP treated film-forming solutions. This indicated the improvement of film formation (smoothness) by HP treatment, which has been previously reported by [109].

##### Ultrasonication

During ultrasound (US) treatment, acoustic waves with a frequency > 16 kHz are transmitted through solid, liquid or gaseous systems. US can be classified into different types based on frequency and intensity [110,111]. As for the modification of potato constituents, US treatment is usually carried out in a liquid medium (usually in a water bath), with starch- or protein-based samples treated in their granular form or in the form of suspensions before or after the gelatinization/gelation process. US can be conducted at room or elevated temperatures. During US treatment cavitation occurs, which is the formation and implosion of gas bubbles that can break polymer chains by mechanical shear forces, which occur when these bubbles collapse. As a side effect, local temperature increases contributing to the modification effect [112]. For different carbohydrate and protein-based films or coatings US treatment has shown to be able to improve gelling properties and tensile properties, increase solubility and surface hydrophobicity, and reduce water vapor permeability [113,114].

For potato starch granules, damaging occurred after US treatment (especially in water [115,116]), which increased with increasing temperature [117], US power [118], US frequency [119], as well as by the application of dual-frequency US-treatment [117]. Furthermore, a decrease in average molecular weight [120] and slight decrease in relative crystallinity [117,118] as well as an increase in amylose content [117] was observed after US treatment. Among these studies, a change in type of crystallinity from B to V type was only reported by Nie et al. [120]. However, other functional properties including thermal, and pasting properties were not significantly influenced by US treatment according to Hu et al. [117], though hydration properties were slightly increased, when compared to native potato starch [115,117].

Mao et al. [121] demonstrated that solubility of potato protein can be increased by US treatment, but only in combination with a pH shift to pH 12. In addition, α-helix content decreased by approx. 10%, while β-sheet content increased by approx. 10%, suggesting a partial unfolding of potato proteins upon US treatment at pH 12. However, Hussain et al. [122] reported an increase in potato protein and patatin solubility by +25% through US treatment at pH 7. Other functional properties including radical scavenging activity, foaming ability, emulsifying ability and emulsifying stability also increased with increasing treatment time for both, potato protein and patatin fraction by two times or more, whereas particle size and foaming stability decreased [122].

In case of potato peel-based films, US treatment can enhance packaging related properties, as tensile properties were increased and hydration properties decreased by US treatment of potato peel-based films [123]. Similarly, an increase in tensile strength and decrease in water vapor permeability was reported by Wang et al. [124] for potato starch-based films after US treatment.

##### Ionizing Irradiation

Upon interaction with ionizing irradiation, cleavage of water molecules (in the product) into free radicals and high energy electrons occur, which can facilitate cross-linking and hydrolysis reaction of the polymer chains. The affinity of a polymer towards crosslinking and chain scission is described by the G(x) and G(s) value (quantification of the chemical yield obtained from ionizing irradiation treatment), respectively, which can highly differ among polymers. Thereby, the ratio between G(s):G(x) indicates the overall prevailing reaction (cross-linking < 1, hydrolysis > 1) which, however, depends on the irradiation dose and temperature. Accordingly, the molecular weight is either increased (cross-linking is prevailing), decreased (chain scission is prevailing) or not significantly changed (cross-linking and chain scission occur equally). For starches (including potato starch) mostly decreasing average molecular weights upon irradiation have been reported in several studies which were reviewed by Bashir and Aggarwal [125], indicating the prevalence of hydrolysis [125]. The induced structural changes by ionizing irradiation can thus alter product properties [126,127,128].

Investigating the effect of electron beam or gamma irradiation on potato starch morphology, none or only slight changes in starch granule surface and color have been observed throughout the reviewed literature [129,130,131]. However, the occurrence of free radicals on starch surface was observed by Rao et al. [129]. A decrease in amylose content and increase in water solubility index with increasing irradiation dose was determined by Atrous et al. [130], who presumed that these findings were attributed to the occurrence of depolymerization hindering the formation of iodine complexes and increasing water affinity. Interestingly, swelling power of potato starch was up to 1.5 times higher when irradiated at ≤20 kGy, similar to native starch when irradiated at 35 kGy (1 times) and significantly lower (0.4 times), when irradiated at 50 kGy [130]. According to the authors [130], this binomial relationship might be attributed to starch granule morphology/degree of disruption, enabling water penetration at first, but then decreasing water holding capability after gelatinization at higher disruption levels.

Overall, gelatinization temperatures, ΔH, viscosity and relative crystallinity of potato starch decreased upon irradiation treatment and with increasing irradiation dose [129]. The decrease in potato starch viscosity by ionizing irradiation treatment and increasing irradiation dose was supported by the findings of Atrous et al. [130] and Teixeira et al. [131].

In potato starch-based films, ionizing irradiation treatment resulted in decreased hydration properties (solubility, swelling, water absorption) [132] and increased hydrophobicity [133]. Investigating the effect of gamma irradiation on tensile properties of potato starch-based films, no change was observed for irradiation treatments at 0–15 kGy [131], but an increase was observed at 30 kGy [133], indicating an irradiation dose dependency. However, Teixeira et al. [132] reported a decrease in tensile properties of electron beam (0–60 kGy) treated potato starch/hibiscus extract-based films, indicating weakening of the films due to depolymerization into shorter chains, upon irradiation [132]. This behavior was in contrast to the findings reported by Cieśla and Sartowska [133] and Teixeira et al. [131], which might be attributed to the different irradiation source (electron beam vs. ^60^CO gamma) and/or a higher amount of incorporated hibiscus extract into potato starch-based films by Teixeira et al. [132].

##### Others

In addition to HP, US treatment and ionizing irradiation, there are other non-thermal modification methods to change the structural and functional properties of potato constituents, such as milling and electric field treatment. 

Milling is commonly applied in the food industry to reduce the particle size and produce flour/powder. Ball-milling of potato starch does not only reduce the particle size of the samples but can also highly damage granule morphology, induce partial gelatinization and destroy B-type crystallinity, which is accelerated with increasing milling time [134]. Furthermore, for jet-milled potato starch a decrease in molecular weight with increasing milling speed was observed [135]. Relative crystallinity, gelatinization temperatures and pasting properties were decreased and thermal stability increased upon jet-milling of potato starch. These effects were gradually accelerated with increasing jet-milling speed [135]. An increase in solubility, swelling and viscosity was observed for high-energy ball-milled potato starch at 0.5–2.2 J/g. Applying higher energy (≥2.8 J/g), a complete loss of crystallinity and viscosity occurred [136]. Based on the results, Juarez-Arellano et al. [136] categorized the effect of ball-milling into three stages: Modification-stage, mechanolysis-stage and over-destruction-stage, depending on the amount of energy applied and the corresponding behavior of the characteristic properties [136].

Electric field treatment is the generic term for a number of different applications, including high voltage electric field (HVEF), or induced electric field (IEF). HVEF application is regarded as a method to physically modify starch properties in the absence of thermal influence, as no significant changes in product temperature occur during this treatment. In HVEF application, the sample is exposed to an electric field in liquid or gaseous medium. High voltage electric field can be classified into high electrostatic field (uniform electric field with no currents or varying voltages) and into high voltage electrical discharge (current flow causing ionizing/plasma) [137]. 

Cao and Gao [138] investigated the effect of HVEF on potato starch properties, by varying treatment time (10–50 min) and dose (10–40 kV). SEM images revealed some granular deformation, which increased with increasing processing time. Compared to native potato starch, apparent amylose content of HVEF treated starches significantly increased with increasing treatment time. Furthermore, relative crystallinity, water solubility and swelling power gradually decreased upon HVEF treatment with increasing voltage and treatment time [138].

In IEF treatment, electromagnetic induction is induced according to Faraday’s law, where energy is conducted from a primary coil to the sample (solution), acting as a secondary coil [139]. In contrast to HVEF, no significant influence of IEF treatment (30 or 96 h) on relative crystallinity of potato starch was observed by Li et al. [140]. However, gelatinization and pasting temperatures increased independent of IEF treatment time and the final viscosity increased by approx. 33% (IEF for 30 h) and 40% (IEF for 96 h). Swelling power of potato starch at 65–95 °C slightly decreased upon all treatment constellations, showing a tendency of decreasing values upon increasing treatment time and temperature [140].

##### Conclusion Physical Modification

In this part of the article, different thermal and non-thermal physical modification methods were reviewed in terms of their influence on the structure and properties of potato constituents. From the evaluated studies, it can be concluded that different physical treatment methods require different preconditions for the sample. For instance, for HMT modification, a granular form of the sample with low moisture content is needed, whereas other methods such as MW treatment or ionizing irradiation can modify samples in form of granules, suspensions/hydrogels or films. Moreover, the required treatment time among different physical modification methods did highly vary, reaching from 5 min for MW treatment up to 24 h or more used for ANN and IEF treatments. Furthermore, different physical modifications generally resulted in different sample properties; the trends are summarized in Table 2. However, it should be considered that qualitative and quantitative changes in sample structure and properties can differ among physical modification methods, due to differences in modification and/or sample parameters (e.g., intensity, temperature, time, moisture content and pH). Especially for potato protein modification, pH is one of the most contributing factors, as the change in some functional properties is only susceptible under acidic or basic conditions.


foods-11-02927-t002_Table 2Table 2General overview of changes induced by different physical modification methods on potato starch, potato protein, and potato-based films structure and properties (specific processing parameters or changes in processing parameters are not considered here, but can be looked up above). Abbrevations and references: **HMT** (heat moisture treatment) [15,29,67,73,74,75], **ANN** (annealing) [20,77,78], **MW** (microwave) [30,68,87,88,90,91], **RF** (radio frequency) [90,92], **FT** (freezing and-thawing) [93,94], **Superheated steam** [97], **HP** (high pressure) [25,100,107,108,109], **US** (ultrasound) [115,116,117,118,119,120,121,122,123,124], **Ionizing irradiation** [129,130,131,132,133], **Milling** [134,135,136], **EF** (electric field) [138,140], **G’** (storage modulus), **WVP** (Water vapor permeability). Meaning of symbols: **↑** (increase), **↓** (decrease), **→** (no sig. influence), **↗** (slight increase), **↘** (slight decrease), **-** (no data available). Meaning of superscript numbers: **1** (for patatin at pH 6), **2** (at pH 7), **3** (at pH 12), **4** (for potato starch-lauric acid-complexes), **5** (for potato starch-hibiscus extract-based films).
Starch
*Modification*
PropertiesStructureCrystallinityHydration propertiesThermal propertiesPasting propertiesDamagingAmylose contentDegreeTypeSolubilitySwelling powerWater absorption capacityGelatinization temperaturesMelting enthalpyPasting temperatureFinal viscosity
*HMT*
↗↑↓B → C/A↑↓↓↑↓↑↓/↑
*ANN*
→-→/↗→↓/↑↓/↑-↑→/↗↑↑
*MW*
↑↓↓/↑B → A↓↑↑↘/→/↗↗→/↑↘/↑
*RF*
↗↓↓B → C---↘↘↘↗
*FT*
↑------↘↘↗↘
*Superheated steam*
→-↓→↓↓-↘↘↑↑
*HP*
↗/↑↗↓/→B → B + V↓↘-↘/→/↗→→↘/→/↗
*US*
↗↑↘→/B → V↗↗↗→→→→
*Ionizing irradiation*
→/↗↓↘→↗↘/↗-↓↓--
*Milling*
↑-↓loss↑↑↑↓-↓↓
*EF*
↗↑↓/→-↓↘/↓-↑→→/↑↑

**Protein**

*Modification*
PropertiesStructureFilm forming propertiesHydration propertiesFoaming propertiesEmulsifying propertiesα-helix contentβ-sheet contentParticle sizeViscositySolubilitySurface hydrophobicityAbility/CapabilityStabilityActivityStability
*Autoclaving*
--
↑→---↗↗
*HP*
↓↑↓-↓ ^1^/→ ^2^/↑↗↗↗-↓
*US*
↓↑↓-→ ^2^/↑ ^2/3^↘↑↓↑↑

**Potato-based films**

*Modification*
PropertiesCrystallinityFilm forming propertiesHydration propertiesThermal propertiesTensile properties
DegreeViscosityG’SmoothnessSolubilityMoisture contentWater absorptionWVPSurface hydrophobicityGlass transition/Thermal stabilityElongation modulusTensile strengthElongation
*HMT*
----↘--↘---↗↗
*HP*
-↑↑↑---------
*US*
↑ ^4^---↓-↓↘ ^4^-↗ ^4^-↑↓ ^4^/↑
*Ionizing irradiation*
-↓--↘-↓-↑--↘/→ ^5^/↗→/↑


### 3.2. Chemical Modification

In this section, the effects of chemical modification on potato constituents are outlined and the most important findings presented in Table 3. In general, chemical modifications refer to the substitution, cross-linking or degradation of a polymer via chemical reaction.

For instance, in substitution reaction, new functional groups/existing functional groups in a polymer are introduced/blocked to alter polymer functionality. In cross-linking modification of polymers new intermolecular linkages are created [141], and in degradation modification of polymers, polymer chains are either cleaved via hydrolysis [142] or functional groups such as hydroxyl groups are oxidized to carbonyl and carboxyl groups [143].

Basically, starch contains a large number of hydroxyl groups, and proteins contain a variety of different functional groups (hydroxyl-, carboxyl-, amine groups, etc.). These functional groups can be used as reactive sides for chemical modification reactions such as acylation, esterification, etherification, cross-linking, grafting, acid hydrolysis and oxidation. A common technique to evaluate chemical changes represents FT-IR analysis [142,144,145,146,147]. Some general information on the chemical modification of starches and proteins can be also found in other review articles [60,148,149,150,151]. In this review article, studies on chemical modifications of potato constituents over the last years are reviewed in detail.

#### 3.2.1. Chemical Substitution

Hydroxyl groups of potato starch and different functional groups of potato proteins can be substituted via chemical reaction. In this review article, substitution reactions are divided into subcategories, dependent on the type of agent used and starch/protein derivate obtained (i.e., acetylation, phosphorylation, fatty acid esterification, octenyl succinate, citric acid esterification and etherification). The amount of derivatization via substitution reaction often influences resulting properties of the product and is commonly described by the degree of substitution, which reflects the amount of theoretically possible substitutions per polymer unit. For instance, the degree of substitution is maximum three for potato starch as each anhydride glucose unit possess three hydroxyl groups, which can be substituted by other groups [152]. Amongst others, the degree of substitution reactions can be adjusted by the acylating agent concentration, treatment time and treatment temperature. For instance, a correlation between the degree of substitution and plasticizer migration was found in starch-based films [153]. Acylation of potato protein with different acylating agents showed that the acylation effect depended on the used acylating agent and reaction temperature, resulting in decreased thermal stability and increased swelling towards water [154]. When acylation agents with multiple functional groups are used, chemical modifications (e.g., phosphorylation, citric acid esterification) can be either used as (single) substitution reaction or as cross-linking reaction, if functional groups of different chains are substituted by the same acylation agent [155,156,157].

##### Acetylation

Acetylation refers to the chemical reaction of potato starch or potato protein with acetic anhydride [158,159,160].

Acetylation of potato starch resulted in an increase in moisture and amylose content as well as increased relative crystallinity, solubility and paste clarity. However, type of crystallinity and gelatinization temperatures of potato starch were not significantly influenced by acetylation [158].

Miedzianka et al. [160] investigated the influence of acetylation on potato protein concentrate and isolate, varying acetylation concentration. Upon acetylation, solubility and water holding capacity of the potato protein concentrate and isolate increased by up to 100%. Foaming properties (capacity and stability) were lowest for potato concentrate and isolate acetylated with the medium concentration of acetic anhydride (1 mL/g). However, regarding the oil binding capacity, emulsion properties and essential amino acid composition, the effect of acetylation on potato protein was highly influenced by protein purity. For instance, upon acetylation the oil binding capacity increased for potato protein concentrate but decreased for potato protein isolate. For potato protein concentrate, the decrease in emulsion activity and increase in emulsion stability was much higher than for potato protein isolate. Upon acetylation, the amount of essential amino acids decreased in potato protein concentrate and isolate. For potato protein concentrates, highest decrease in essential amino acids was determined at the lowest acetylating agent concentration (0.4 mL/g) and for potato protein isolates, highest decrease in essential amino acids was measured at the highest acetic anhydride concentration (2 mL/g) [160].

##### Phosphorylation

Phosphorylation of starch can be performed using different agents including phosphoryl chloride, sodium trimetaphosphate, sodium tripolyphosphate and sodium or potassium orthophosphate to obtain mono- and/or di-starch phosphates. In di-starch phosphates, orthophosphate groups can act as intramolecular or intermolecular bridges between the C2, C3 and C6 atoms of glucose units from the same or different (cross-linking) chain (see Section 3.2.2) [155].

Phosphorylation of potato starch resulted in slightly decreased moisture content, no change in amylose content and a slight change in color, which was however not visible to the human eye. Compared to native starch, gelatinization temperatures were slightly decreased when phosphorylation was performed at 15 °C, but slightly increased when phosphorylation was performed at 45 °C. The authors interpreted these effects as loosening and strengthening of the potato starch structure at 15 °C and 45 °C, respectively. Accordingly, an increase in viscosity was observed in potato starch pastes, which were phosphorylated at 45 °C [155].

For the phosphorylation of potato proteins, phosphorus oxychloride, phosphoric acid and sodium trimetaphosphate can be used [161]. Miedzianka and Pęksa [161] phosphorylated potato protein at different pH values (5.2, 6.2, 8.0 and 10.5) using sodium trimetaphosphate as the phosphorylating reagent. Properties of potato protein were greatly affected by phosphorylation under alkaline conditions (pH 8.0) showing the most significant increase in water absorption capacity (approx. 20%), oil absorption capacity (2.5 times), emulsifying activity (approx. four times) and foaming capacity (more than two times), when compared to native potato protein and potato protein modified at other pH values. However, protein solubility was not significantly affected by phosphorylation [161]. 

##### Fatty Acid Esterification

Another approach to modify potato starch by chemical substitution is the esterification of hydroxyl groups by fatty acid chains. Vanmarcke et al. [144] investigated the effect of fatty acid chain length (C8, C12 and C16) on potato starch fatty acid ester cast films. Results showed that chain length did not influence esterification reactivity and FT-IR spectra confirmed the almost complete substitution of hydroxyl groups by disappearance of the 3300 cm^−1^ band. However, with increasing chain length, weight of the samples increased and the X-ray diffraction peak in the low angle region shifted towards lower angles, indicating an increase in nanometer scale ordered structure based on the esterification of longer fatty acid chains. Thermographs showed that the thermal stability of potato starch increased due to fatty acid esterification, as hydrophobicity increased, which was indicated by the fact that the mass did not decrease due to water evaporation and the starch degradation shifted to higher temperatures. However, no significant differences between different chain lengths were observed. Fatty acid chain length highly influenced tensile properties of the modified starch films. With increasing chain length from C8, C12 to C16, elongation at break decreased by 150%, 133% and 13%, respectively. Tensile strength and elongation modulus showed no linear trend but highly depended on the fatty acid chain length. For C8, C12 and C16 fatty acid esterified potato starch films, the tensile strength was measured to be 3.8, 2.5 and 6.5 MPa and for the elongation modulus 68, 40 and 122 MPa was determined, respectively [144].

##### Octenyl Succinylation

The hydrophobic group of alkenyl succinic anhydrides can be esterified with starch, to weaken internal bonding and disorder the internal structure to obtain more fluid and clear pastes. Due to its food approval at low substitution levels, the influence of octenyl succinic anhydride on starch properties is widely studied [162,163,164].

Wang et al. [162] demonstrated that potato starch granule size has an effect on the efficiency of octenyl succinylation. For octenyl succinic anhydride modified starch, higher degree of substitution (DS) was reported with decreasing granule size varying between 0.0147–0.0219 (degree of substitution), indicating that smaller particles with higher surface areas are more susceptible to modification. However, this was only observed when octenyl succinic anhydride modification was carried out prior to particle fractioning [162].

Investigating the effect of octenyl succinic anhydride modification on potato starch structure, Won et al. [164] observed an increase in surface roughness, porosity, granule size and the occurrence of cavities and slight deformation after octenyl succinic anhydride modification. However, no differences between different degrees of substitutions were observed and crystallinity of potato starch was not affected by octenyl succinic anhydride modification [164].

Compared to native potato starch, octenyl succinic anhydride modified starch showed lower amylose leaching, gel strength and viscosity, which gradually decreased with increasing degree of substitution. Won et al. [163] explained these results by hindered interactions between amylose chains caused by octenyl succinic anhydride modification, which resulted in decreased amylose leaching, retrogradation and therefore weaker gel formation. The weakening of inter- and intramolecular bonding was also reflected by the decrease in final viscosity. However, gelatinization temperatures were not influenced by octenyl succinic anhydride modification [163].

In another study, slightly increased amylose content, swelling power, gelatinization temperatures and slightly decreased water binding capacity and pasting temperature were observed upon octenyl succinylation of potato starch [165].

##### Citric Acid Esterification

Based on its three carboxyl groups and acidity, it has been reported that citric acid can induce esterification, cross-linking and hydrolysis, when reacting under elevated temperatures with potato starch [156,157].

SEM images of citric acid esterified potato starch revealed the occurrence of some surface damage due to citric acid treatment, which was, however, not intensified using a higher citric acid concentration (up to 10%). Furthermore, amylose content of potato starch slightly in-/or decreased and moisture content and relative crystallinity decreased upon citric acid modification [156,166]. A change in type of crystallinity was not reported [156]. In addition to structural changes, citric acid esterification of potato starch also influenced different functional/physicochemical properties. Upon citric acid esterification, the solubility of potato starch decreased, whereas swelling power, water binding capacity and gelatinization temperatures increased. The reported effects of citric acid modification on potato starch pasting properties were slightly different, although pasting temperature did not appear to be significantly affected and final viscosity appeared to increase with citric acid modification compared to native potato starch [156,166]. In addition, Kapelko-Żeberska et al. [167] showed that increasing processing temperatures increased the degree of substitution as well. Accordingly, decreased solubility, swelling power and gelatinization temperatures were observed upon citric acid modification that further deceased with increasing processing temperature. 

The effect of citric acid modification (5–20% *w*/*w* potato starch) to reduce hydrophilic properties of potato starch/chitosan composite films was studied by Wu et al. [146]. Cross-linking of composite films resulted in decreased smoothness but also decreased hydration properties (moisture content, water solubility, and swelling degree) as well as decreased water vapor permeability and moisture sorption. Furthermore, mechanical properties (tensile strength and elongation at break) and thermal properties increased after cross-linking reaction. All changes in film properties gradually increased with increasing citric acid concentration up to 15%. Excessive use of citric acid (20%) led to solidification and crystal formation on the film surface.

##### Etherification

The etherification reaction of potato starch with various reagents, including those containing carboxymethyl, hydroxypropyl and/or hydroxyethyl groups, requires alkaline catalysts, of which sodium hydroxide (NaOH) is usually used. A common etherification reaction for starches is the hydroxypropylation, which has been extensively reviewed elsewhere [168]. 

In the reviewed literature, carboxylation [169], hydroxypropylation [145] and hydroxyethylation [170] of potato starch were found to alter product structure and properties. In addition to reagent concentration, other reaction and solvent related variables such as reaction temperature, reaction time, catalyst concentration, type of solvent and solvent concentration were found to affect the degree of substitution and the product properties [169]. Therefore, optimal etherification conditions need to be determined individually, as demonstrated by Prajapati et al. [169] for carboxypropylated potato starch. However, in general etherification of potato starch resulted in granular surface damaging [170], increased solubility [145], higher final viscosity [170], decreased crystallinity [145], decreased pasting temperatures [145] and decreased thermal stability [170]. Similar to other substitution reactions such as citric acid esterification, properties of potato starch were found to decline when optimal reaction conditions are exceeded [170].

#### 3.2.2. Chemical Cross-Linking

Cross-linking of potato starch describes the formation of intermolecular covalent bonds (between different chains) via esterification or etherification reaction with a cross-linking agent. Thereby, the cross-linking agents possess bi- or multi- functional groups, which enable the reaction on different sides with multiple chains.

In the case of starch, it has been reported that cross-linking only occurs between two amylopectin molecules or between amylopectin and amylose, but not between two amylose molecules [171]. The reduction in amylose content by cross-linking reaction, as cross-linked molecules can be considered as amylopectin molecules, can (among others) be used to determine the degree of cross-linking by the starch–iodine method [171]. 

Cross-linking of potato starch can be performed using cross-linking agents such as acetylmalic acid chloroanhydride [172], sodium trimetaphosphate (STMP)/sodium tripolyphosphate (STPP) [173] and deep eutectic solvents [174]. According to Shulga et al. [172] cross-linking of potato starch resulted in destruction of its granular form and reduced relative crystallinity as well as reduced thermal stability. Furthermore, Heo et al. [173] reported that the increased formation of covalent bridges between starch molecules due to cross-linking hindered granular swelling and thermally induced gelatinization. In addition, pasting temperature and (final) viscosity of potato starch pastes increased upon cross-linking.

For potato starch-based films, the formation of new covalent bonds through cross-linking generally strengthened the starch gel network, resulting in an increase in rheological, thermal and tensile (elongation modulus, tensile strength) properties. Solubility of potato starch-based films was not affected and water sorption degree increased upon deep eutectic solvent addition [174].

#### 3.2.3. Degradation

##### Acid Hydrolysis

Acid hydrolysis describes the process of chemical bond cleavage via a nucleophilic substitution reaction with water under thermal conditions and in an acidic environment. Common acids that are used in acid hydrolysis include hydrochloric acid (HCl), acetic acid, citric acid and sulfuric acid (H_2_SO_4_).

Generally, hydrolysis of potato starch in water results in a decrease in molecular weight and some granular damage. Furthermore, an increase in thermal properties (gelatinization temperatures and stability), gel strength and viscosity of hydrolyzed potato starch was observed [142,166,175].

Reviewing several studies, it seems that the qualitative and quantitative impact of acid hydrolysis on potato starch did not only depend on the amount but also on the type of acid as well as the type of solvent used [142,175]. For instance, Celikci et al. [142] showed that 0.25–1.25 M of HCl or H_2_SO_4_ influenced the quantitative increase in bonding strength of potato starch-based adhesives. However, when alcohol–water mixtures were used as a solvent instead of water, bonding strength of the hydrolyzed starch-based adhesives decreased, when compared to the native reference [142]. Furthermore, the use of water as solvent led to surface cracking, an increase in thermal properties and to a slight decrease in relative crystallinity, whereas the use of alcohol or alcohol–water mixtures resulted in the occurrence of granular pitting, a decrease in thermal properties, no change in relative crystallinity and an increase in solubility [175].

Varying the amount of acid used for potato starch hydrolysis, the degree of hydrolysis is usually affected, which can in turn also influence qualitative and quantitative changes in potato starch properties. On the one hand, a lower degree of hydrolysis, (using milder conditions) seem to accelerate changes in potato starch including a further decrease in amylose content and hydration properties, and increase in thermal stability and viscosity [142,166]. On the other hand, the use of high acid concentrations (≥1 M) can result in opposite effects such as a decrease in viscosity [142]. 

A favorable effect of mild acid hydrolysis was also observed for film formation of potato peel-based films by Merino et al. [176] as the films became smoother, more homogeneous, more transparent and more flexible after acid hydrolysis treatment (1 M acetic acid), which was performed prior to cast film plasticization. 

##### Oxidation

Oxidation of potato starch can be carried out using oxidizing agents under controlled pH to oxidize hydroxyl groups into carbonyl and carboxyl groups. Common oxidizing agents used, include sodium hypochlorite, hydrogen peroxide and ozone [143]. Degree of oxidation is usually expressed as the sum of carbonyl and carboxyl content. In contrast to the use of chemical oxidizing agents, ozone can be decomposed after the oxidation reaction into oxygen by an ozone destructor. Thus, ozone oxidation is often described as a safe and environmental friendly method [143], which can also be applied to fresh potato tubers to reduce the spread of potato diseases during storage [177]. 

Basically, it was found that with increasing concentration of oxidizing agent and increasing treatment time, the degree of oxidation of oxidized potato starch increased [147,178,179] and the amylose content decreased, indicating depolymerization [179]. Higher degrees of oxidation, which means the conversion of hydroxyl groups into carbonyl and carboxyl groups, resulted in increasing granule surface cracking and shape deformation and in a decrease in Maltese cross intensity, determined by polarized light microscopy [178,179]. B-type crystallinity of potato starch was not affected by oxidation [179], but change in relative crystallinity varied upon the amount of oxidant agent concentration used [178]. Authors suggested that oxidation occurred first in the amorphous (increasing crystallinity at low oxidant concentration), and second in the crystalline structures (decreasing crystallinity at higher oxidant concentrations) [178]. However, no change in relative crystallinity was reported by Castanha et al. [179] for ozonized potato starch, but particle size was slightly decreased [180]. Overall, an increase in solubility, decrease in swelling power, increase in pasting temperature and decrease in final viscosity upon potato starch oxidation was observed in several studies [178,179]. The induced changes were amplified with increasing oxidant agent concentration and treatment time [178,179]. 

Oxidation treatment of potato starch prior cast film formation revealed positive effects on the potato starch films such as improved smoothness and transparency [181]. B-type crystallinity of potato starch-based films was not affected by oxidation [182] but relative crystallinity of films plasticized with glycerol or sorbitol decreased according to Fonseca et al. [182] using sodium hypochlorite, whereas it decreased for glycerol plasticized films upon ozone treatment [181]. Similarly, no change in moisture content or water vapor permeability and slight increase in solubility were observed in ozonized films [181], but slightly decreased values were reported by Fonseca et al. [182]. Furthermore, according to La Fuente et al. [181] elongation at break of potato starch-based films decreased and elongation modulus increased upon oxidation. Similar results were found by Fonseca et al. [183] who reported a decrease in elongation at break upon oxidation but an oxidant agent concentration dependency of changes in elongation modulus and tensile strength of oxidized potato starch-based films. In a follow-up study, the high influence of plasticizer type and amount used in potato starch-based films on the qualitative and quantitative changes in tensile properties induced by oxidation was demonstrated [182]. Overall, the different effects on different film-related properties such as hydration properties, water vapor permeability and tensile properties were determined for potato starch-based films upon oxidation which might be attributed to the different oxidizing agents and/or plasticizer (concentrations) used [181,182].

#### 3.2.4. Conclusion Chemical Modification

Based on the reviewed literature, several substitution, cross-linking and degradation reactions have been studied to chemically modify potato constituents. In most cases potato starch was analyzed. Overall, chemical modifications can change physicochemical and functional properties as summarized in Table 3. Through chemical modification, constituents obtain different functional groups depending on the degree of substitution and change their molecular weight by either cross-linking, hydrolysis or oxidation.

There are qualitative and quantitative differences in sample structure and properties depending on the chemical modification method used. Parameters that influence the method are e.g., treatment time, temperature, type and amount of chemical(s), while e.g., solvent, purity, moisture content and pH-value influence the raw material, i.e., the potato constituents. As an example, the purity of the potato protein (either concentrate or isolate) has shown to not only quantitatively, but also qualitatively influence hydration and emulsifying properties upon chemical modification.


foods-11-02927-t003_Table 3Table 3General overview of changes induced by different chemical modification methods on potato starch, potato protein and potato-based films structure and properties (specific processing parameters or changes in processing parameters are not considered here, but can be looked up above). Abbreviations and references: **Acetylation** [158,160], **Phosphorylation** [155,161], **FA** (fatty acid) **esterification** [144], **OSA** (octenyl succinic anhydride) **modification** [162,163,164,165], **Citric acid esterification** [146,156,166,167], **Etherification** [145,169,170], **Cross-linking** [172,173,174], **Acid hydrolysis** [142,166,175,176], **Oxidation** [147,178,179,181,182,183], **Acylation** [154], **G’** (storage modulus), **WVP** (Water vapor permeability). Meaning of symbols: **↑** (increase), **↓** (decrease), **→** (no sig. influence), **↗** (slight increase), **↘** (slight decrease), **-** (no data available).
Starch
*Modification*
PropertiesStructureCrystallinityFilm forming propertiesHydration propertiesThermal propertiesPasting properties
DamagingAmylose contentDegreeTypeViscositySolubilitySwelling powerWater absorption capacityMoisture contentThermal stabilityGelatinization temperaturesMelting enthalpyPasting temp.Final viscosity
*Acetylation*
-↑↑→-↑--↑-→↓--
*Phosphorylation*
-→--↑---↘-↘/→/↗→/↗--
*FA esterification*
---------↑----
*OSA modification*
↗↗-→↓-↗↘--→/↗↘/→↘↓
*Citric acid esterification*
↗↘/↗↓→-↓↗↗↓-↗↓↘/→↘/↑
*Etherification*
↗-↓--↑---↓--↓↑
*Cross-linking*
↗/↑-↓-↑-↓--↓--↗↑
*Acid hydrolysis*
↑↓↘→↓/↑↑↓-↓↑↘/↑↓→↑
*Oxidation*
→/↗↓↓/→/↑→-↑↓↓----↗↓

**Protein**

*Modification*
PropertiesStructureHydration propertiesThermal propertiesFoaming propertiesEmulsifying properties
α-helixcontentβ-sheet contentSolubilityWater absorption capacityOil binding/absorption capacityThermal stabilityAbility/CapabilityStabilityActivityStability
*Acylation*
---↑-↓----
*Acetylation*
--↗/↑↑↘/↗-↘↘/↗↓/↘↑/→
*Phosphorylation*
--→↗↑-↑-↑-

**Potato-based films**

*Modification*
PropertiesCrystallinityFilm forming propertiesHydration propertiesThermal propertiesTensile properties
DegreeViscosityG’SmoothnessSolubilityMoisture contentWater absorptionWVPSurface hydrophobicityGlass transition/Thermal stabilityElongation modulusTensile strengthElongation
*Citric acid esterification*
---↘↓↓↓↓↑↑-↑↑
*Cross-linking*
-↑↑-→-↑--↑↑↑↓/→
*Acid hydrolysis*
---↑---------
*Oxidation*
↘/↑---↓/→/↗--↘/→↑-↓/↑↘/→/↗↓


### 3.3. Biochemical Modification

In general, biochemical modifications of starches and proteins including enzymatic substitution (Section 3.3.1), cross-linking (Section 3.3.2) or hydrolysis (Section 3.3.4) are usually regarded as a clean or green alternative to chemical modification (Section 3.2). Throughout the different biochemical modification methods, substrate specific enzymes can be used such as in enzymatic de-/branching modification (Section 3.3.3), where the branched structure of potato starch can be altered to effect starch crystallinity and thus its properties.

#### 3.3.1. Enzymatic Substitution

Substitution of hydroxyl groups of potato starch building blocks can be either performed by chemical substitution as outlined in Section 3.2.1 or by enzymatic modification. One example of enzymatic substitution is the esterification/transesterification of fatty acids by lipases. Based on fatty acid solubility, organic solvents or ionic liquids need to be used. Dependent on the lipase used, degree of esterification and esterification site can vary [184].

According to Zarski et al. [185] degree of substitution gradually decreased with increasing reaction temperature (60–80 °C) and time (4–8 h), when enzymatic potato starch esterification with oleic acid by lipase in an ionic liquid was performed. The authors reported that a longer reaction time resulted in an increase in water formation, which changed pH, increased hydrolysis and possibly decreased lipase activity. The results showed that esterification of potato starch with oleic acid led to a loss in crystallinity and SEM images revealed the destruction of granule shape and the smooth surface due to the enzymatic treatment. The authors suggested that these results are due to hydrogen bond disruption and substitution by ionic liquids and esterification. A decrease in thermal stability due to enzymatic esterification was reported in terms of the initial and onset degradation temperatures, which decreased from 251.8 and 300.1 °C to 198.4 and 150.7 °C, respectively [185].

#### 3.3.2. Enzymatic Cross-Linking

Enzymatic cross-linking of potato protein can be performed in a variety of different ways including acyl-transfer reaction, oxidation/radical formation and 1,4-addition reaction. For potato protein, enzymatic cross-linking using transglutaminase, laccase, tyrosinase and peroxidase has been reported [186,187,188,189].

Of these four enzymes, transglutaminase and peroxidase showed higher quantitative impacts on potato protein properties than tyrosinase and laccase [189]. For instance, thermal stability of potato protein was increased to a higher extent using transglutaminase and peroxidase than using laccase or tyrosinase as the cross-linking enzyme. Similarly, rheological properties increased using transglutaminase and peroxidase but decreased using laccase or tyrosinase as the cross-linking enzyme [189,190,191]. The extent of enzymatic cross-linking using laccase could be enhanced using a mediator such as ferulic acid [190]. Furthermore, Glusac et al. [192] demonstrated that tyrosinase-cross-linking could improve emulsion properties of potato protein.

Upon enzymatic cross-linking, an increase in structural order of potato protein was observed in different studies due to a decrease in random coil content [189] and increase in β-sheet content as well as relative crystallinity [189,191]. In a follow-up study, Gui et al. [193] demonstrated that the combination of potato starch and enzymatically cross-linked potato protein resulted in increased intermolecular interactions in the modified mixture, but in decreased intermolecular interactions in the native mixture. In turn, the increase in molecular interactions enhanced gel formation, pasting and rheological properties [192,193], which was also observed in cross-linked potato flour [191].

#### 3.3.3. Enzymatic De-/Branching

During enzymatic branching of potato starch, α-1,4-glycosidic linkages are cleaved and new α-1,6-glycosidic bonds are formed via a transglycosylation reaction using enzymes such as branching enzymes and/or transglucosidase. Therefore, polymer chain length distributions shift from longer chains towards shorter chains, indicating the occurrence of enzymatic hydrolysis. This means that starch polymers become more branched, resulting in a decrease in amylose and increase in amylopectin content [194,195]. Overall, branching of potato starch induced granular damaging, affected type of crystallinity (B → B + C or C), decreased relative crystallinity, gelatinization temperatures and shear viscosity and increased solubility [195]. 

The opposite happens during enzymatic debranching of potato starch, where debranching enzymes such as pullulanase [194] or isoamylase [196] can be used. Here α-1,6-glycosidic bonds are cleaved and new α-1,4-glycosidic bonds are formed. As a result, starch polymers become less branched, resulting in a decrease in amylopectin and increase in amylose content [196]. Accordingly, average molecular weight of the samples gradually decreased with increasing isoamylase concentration and granular damaging occurred, which intensified with increasing enzyme concentration. However, B-type crystallinity, thermal properties and solubility were not significantly affected upon isoamylase treatment, but gel strength and viscosity of debranched potato starch pastes decreased [196]. 

Investigating the influence of debranching on potato starch–lauric acid-based films, increasing debranching time up to 1.5 h gradually decreased surface roughness, water vapor permeability and elongation of the films, whereas tensile strength gradually increased. However, a longer debranching time (2 h) resulted in a reverse effect on potato starch–lauric acid-based films properties [194].

#### 3.3.4. Enzymatic Hydrolysis

Enzymatic hydrolysis of potato starch and potato protein refers to the enzymatic cleavage of glycosidic and peptide bonds, respectively, to decrease chain length and molecular weight and therefore, alter product properties.

Different enzymes including α-amylase [197,198,199], glucoamylase [198], glycosyltransferase [198], or mixtures of different enzymes [199] can be used to hydrolyze potato starch. In general, the enzymatic hydrolysis of potato starch results in a decrease in average molecular weight, average chain length and therefore amylose content [197,198,199]. In starch granules, this was expressed by surface roughness and cracking, which increased with increasing enzyme concentration used, which in turn increased the degree of hydrolysis [197].

Furthermore, enzymatic potato starch hydrolysis resulted in increasing water solubility and absorption capability and in a decrease in gel strength and (final) viscosity [197,199]. No changes in moisture content [199], or pasting temperature [197] were observed upon enzymatic hydrolysis. However, there were differences reported by Asiri et al. [197] and Vafina et al. [199] regarding the changes in thermal properties of potato starch induced by enzymatic hydrolysis using different or similar enzymes. While Asiri et al. [197] did not observe any significant influence of hydrolysis using α-amylase on gelatinization temperatures of potato starch, Vafina et al. [199] reported a decrease in gelatinization temperature and thermal stability upon hydrolysis using a commercial enzyme-mixture or α-amylase. 

For potato protein, the influence of hydrolysis time and different enzymes including protease and alcalase was investigated in several studies [200,201,202]. Upon enzymatic hydrolysis, secondary structure of potato protein was affected, as the amount of α-helix increased and β-sheet content decreased with increasing enzyme concentration [200]. In all studies, degree of hydrolysis gradually increased with increasing reaction time [200,201,202]. DSC analysis performed by Galves et al. [201] indicated that protease treatment of potato protein causes protein denaturation, as the endothermic peak correlated with protein denaturation disappears in native potato protein samples. According to Akbari et al. [200] the number of carboxyl and amino groups increased upon enzymatic hydrolysis, as molecular weight decreased, which in turn increased protein–water interactions. This was expressed in an increase in protein solubility. According to the authors [200], the observed increase in foam capacity and decrease in foam stability with increasing degree of hydrolysis could be related to the increase in solubility enabling shorter protein chains to migrate to air bubble interfaces for faster stabilization, but decrease in protein–protein interactions, which are necessary for long-term stabilization against environmental conditions [200]. Similar changes in protein solubility and foaming properties were also found by Miedzianka et al. [202]. In addition increased emulsifying properties of potato protein were observed upon enzymatic hydrolysis [200].

#### 3.3.5. Conclusion Biochemical Modification

Changes in potato starch structure and properties by enzymatic substitution, hydrolysis, branching and debranching are depicted in Table 4. Furthermore, Table 4 outlines the changes in structure and properties of potato protein, induced by enzymatic hydrolysis and cross-linking, and changes in potato starch-based films induced by debranching. Overall, reviewed biochemical modifications are highly pH and temperature specific, as the used enzymes require different optimal conditions. Compared to chemical modifications, enzymatic modifications often require more time to achieve a similar outcome. However, enzymatic treatment resulted in a high impact on starch and protein morphology and structure, causing changes in crystallinity, as well as pasting, thermal, hydration, foaming and emulsion properties, which depend on the type of enzyme, enzyme concentration and treatment time used.

### 3.4. Dual Modification

Dual modifications are combinations of different physical, chemical and/or biochemical modifications, which are applied simultaneously or successively.

#### 3.4.1. Physical–Physical

While some studies extensively investigated the relationship between modification parameters and the corresponding effect on structural and functional properties of potato constituents applying a certain type of modification, other studies investigated the effect of combined modification in comparison to native samples and/or the effect of an individual modification. Common dual modifications include the combination of thermal treatments such as HMT, ANN, MW, FT with non-thermal treatments such as HP, US and milling [203,204,205,206], but different heating methods have been combined as well [207,208]. 

As demonstrated by Wang et al. [203] and Cao and Gao [209], the order in which the two different physical treatment methods are performed can also influence the resulting qualitative and quantitative properties of the dual-modified samples, compared to native samples. For instance, ANN treatment prior to HP treatment resulted in a decrease in potato starch relative crystallinity, whereas an increase was reported when physical treatments were applied vice versa [203]. In potato starch, solubility and swelling power decreased, and gel hardness increased, when treated with US prior to electric field treatment, compared to untreated native potato starch. Treating potato starch first with an electric field and then with US resulted in reverse effects meaning increase in solubility and swelling power. Interestingly, the simultaneous treatment of electric field and US resulted in similar changes as the electric field prior to US treatment, except for a decrease in swelling power and increase in gel adhesiveness [209]. Some additional general information on dual-modification of starch can be found elsewhere [210].

#### 3.4.2. Physical-Chemical

To enhance the effect of chemical modification on product properties, several studies were performed in the last years, combining different physical treatment methods with chemical modification methods. For instance, the assistance of acetylation or octenyl succinylation of potato starch by high voltage electric field, ultrasonication, pulsed electric field or microwave treatment was reported to increase the degree of substitution compared to a single chemical modification [211,212,213,214]. Moreover, some structural, pasting, rheological, thermal and other functional properties were affected by US treatment assisted acid hydrolysis [215], dry heat and CaCl_2_ dual-treated [216] and annealing treated acetylated [217] potato starch samples.

#### 3.4.3. Physical–Biochemical

Among biochemical modifications, enzymatic hydrolysis is often reported to be amplified by a physical pre-treatment. For instance, enzymatic hydrolysis of potato protein was increased by US treatment [218], and the enzymatic hydrolysis of potato starch by HP [219] and HMT [220]. This can be mostly attributed to an increase in susceptibility due to increasing structural damaging and/or surface cracking. With increasing degree of hydrolysis, different sample properties can be further in-/decreased, as demonstrated by Mu et al. [219] with increasing enzyme concentration and pressure. 

#### 3.4.4. Chemical–Chemical

Similar to dual physical modification, which has been described above (Section 3.4.1), dual chemical modification can be performed as well. For potato starch different dual modifications, including etherification + esterification, etherification + acid hydrolysis, and CaCl_2_ treatment + succinylation, have been reported [221,222,223]. One commonly applied combination is acetylation + cross-linking. In potato starch granules, this dual modification resulted in a decrease in granule size and relative crystallinity and in an increase in thermal and pasting properties. Regarding potato starch-based films, acetylation prior to cross-linking caused lower water vapor permeability, solubility, moisture sorption and relative crystallinity, while flexibility of the films was increased [224].

#### 3.4.5. Chemical–Biochemical

No combined treatment of chemical and biochemical modification of potato constituents was found by the authors during the literature search.

#### 3.4.6. Biochemical–Biochemical

Dual biochemical modification can accelerate the impact on potato constituent properties, compared to the single treatments. As reviewed above, the combined use of branching enzymes and transglucosidase had a synergistic effect that more strongly influenced the properties of potato starch [195].

## 4. Overall Conclusions

In this review article, various physical, chemical and biochemical modifications of potato constituents were identified and the resulting structural and property changes were presented in the text as well as qualitatively illustrated in Table 2, Table 3 and Table 4. Overall, most of the scientific literature on physical and chemical modification was found for potato starch, and only a few selective studies on potato protein, potato flour or potato peel. Regarding biochemical modification, most studies focused on potato protein modification, followed by potato starch. This indicates a contained research interest on physical, chemical and biochemical modification of other potato constituents besides starch. This relates especially to potato protein, potato starch–protein interactions and bio-based plastics derived from different potato constituents, as some of the studies suggested synergistic effects on potato starch–protein interactions and resulting functional properties. As discussed earlier, the need for high value utilization of potato constituents, in form of isolated starch, proteins or unpurified side streams, is based on the large amounts of side streams occurring along the fresh and processed potato supply chain. However, it became evident that modification of potato constituents is highly complex, due to the great influence of processing parameters (including temperature, time, solvent, concentration and pH) and material properties (including chemical composition and moisture content) resulting in quantitative and qualitative differences throughout modifications. For enhancing the use of potato constituents also for other applications than for food products, modifications are required. For instance, using modified potato constituents can lead to films applicable for packaging applications, as they can have a positive effect on film forming, hydration, barrier and mechanical properties.

## Figures and Tables

**Figure 1 foods-11-02927-f001:**
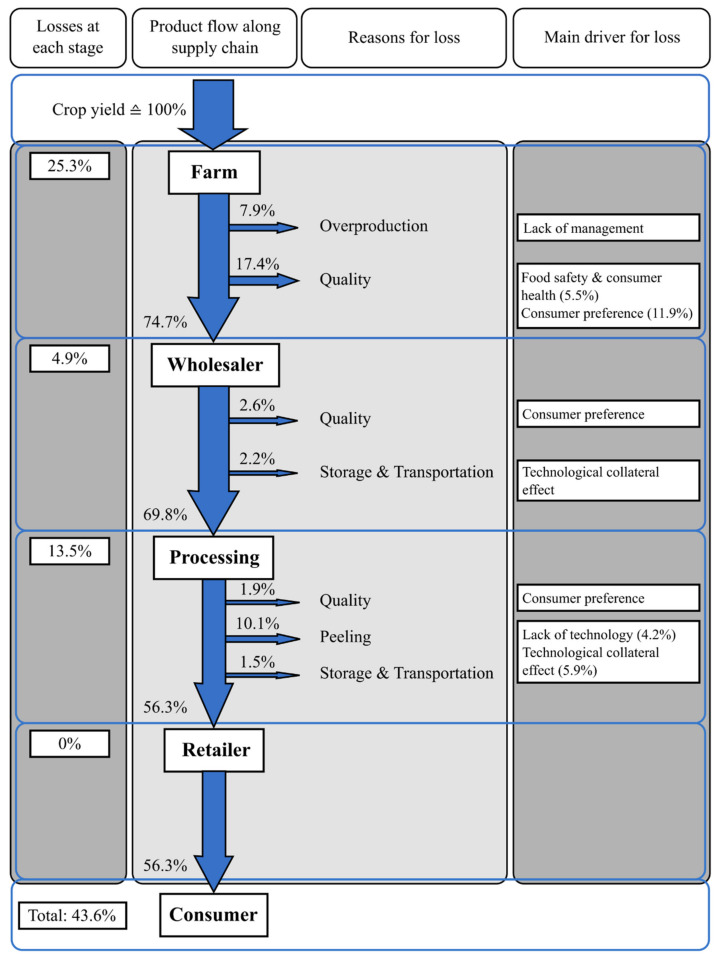
Mean food loss rates of non-organic processing potatoes in Switzerland at each stage of the total agricultural potato production, total supply chain losses and cause of losses. All percentages are related to an initial crop yield of 100%. ↓ Thickness of arrows represent percentage of total product flow. Cause of loss “quality” refers to potatoes, which did not suffice according to the standard quality specifications of the ‘‘Swiss trade customs for potatoes”. Cause of driver “Food safety & consumer health” refers to green and rotten potatoes, and “Consumer preference” refers to “Consumer preferences for certain aesthetic standards or typologies of food” (own illustration based on data published by Willersinn et al. [4]).

**Figure 2 foods-11-02927-f002:**
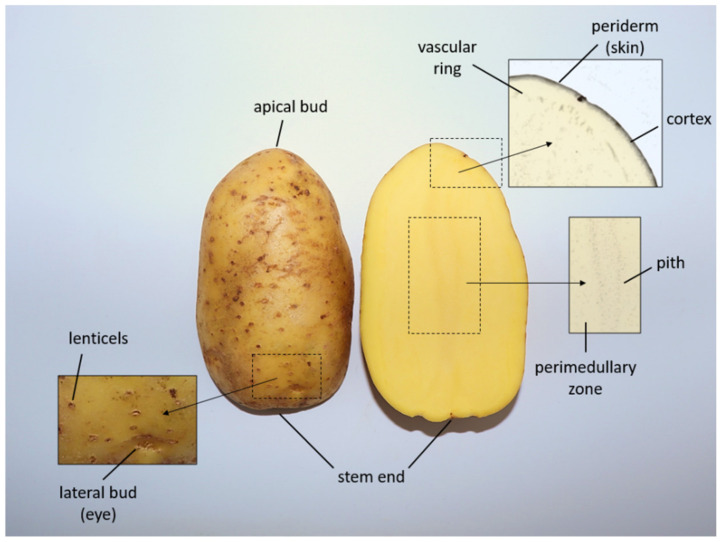
Structure of a potato tuber of the bernina variety (own illustration, terminology based on [28]).

**Figure 3 foods-11-02927-f003:**
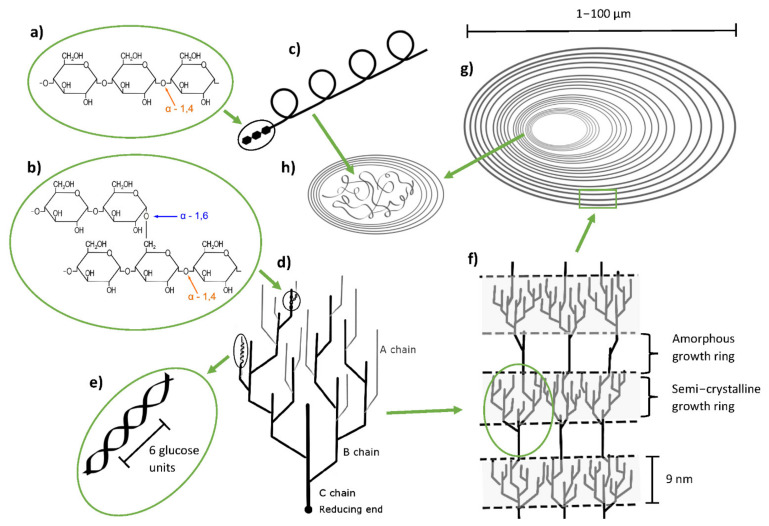
Schematic illustration of some key elements of the macro- and microstructure of potato starch (**a**) molecular structure of amylose, (**b**) molecular structure of amylopectin, (**c**) helical arrangement of amylose, (**d**) arrangement of amylopectin, (**e**) double helix structure of neighboring A and B chains, (**f**) cluster model of amylopectin with alternating amorphous and semi-crystalline parts, (**g**) potato starch granule, (**h**) starch hilum containing mainly amylose. (Own illustration, based on references [37,38,39]).

**Figure 4 foods-11-02927-f004:**
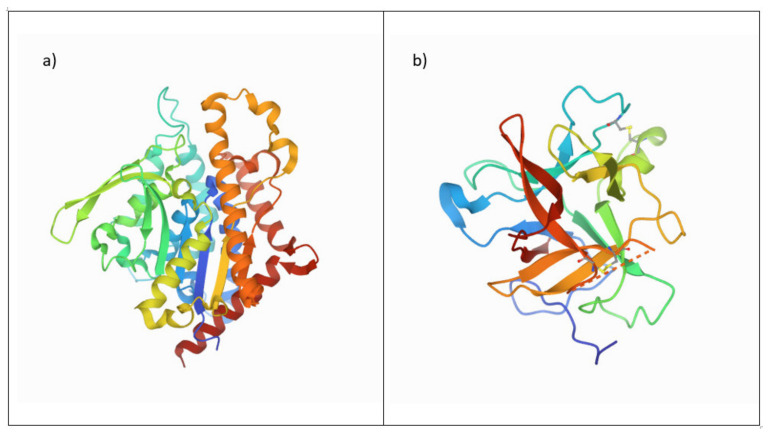
Images from the RCSB PDB (rcsb.org) of the tertiary structures of (**a**) native patatin, PDB ID 4PK9 [58] and (**b**) serine protease inhibitor, PBD ID 3TC2 [59], reprinted with permission from Ref. [59]. 2012, International Union of Crystallography.

**Table 1 foods-11-02927-t001:** Amino acid composition of potato protein with the reactive groups of their side chains, ordered from the smallest to biggest fraction (adopted and extended from [39,61]).

Amino Acid	Abbreviation	Range (%)	Reactive Groups
Cysteine	Cys	0.2–1.3	Sulfhydryl
Tryptophan	Try	0.3–1.9	Indole
Methionine	Met	1.2–2.25	Thioester
Histidine	His	2.0–2.5	Imidazole
Isoleucine	Iso	3.7–5.8	
Glycine	Gly	4.3–6.1	
Tyrosine	Tyr	4.5–5.7	Hydroxyl
Threonine	Thr	4.6–6.5	Hydroxyl
Alanine	Ala	4.6–5.6	
Proline	Pro	4.7–5.6	
Arginine	Arg	4.7–5.7	Guanidino
Phenylalanine	Phe	4.8–6.5	
Valine	Val	4.9–7.4	
Serine	Ser	4.9–5.9	Hydroxyl
Lysine	Lys	6.7–10.1	Amino
Leucine	Leu	9.6–10.7	
Glutamic acid	Glu	9.6–11.8	Carboxyl
Aspartic acid	Asp	11.7–13.9	Carboxyl

**Table 4 foods-11-02927-t004:** General overview of changes induced by different biochemical modification methods on potato starch, potato protein and potato-based films structure and properties (specific processing parameters or changes in processing parameters are not considered here, but can be looked up above). Abbrevations and references: Substitution [185], Branching [195], Debranching [194,196], Hydrolysis [197,198,199,200,201,202], Cross-linking [189,190,191,192,193], **G’** (storage modulus), **WVP** (Water vapor permeability). Meaning of symbols: **↑** (increase), **↓** (decrease), **→** (no sig. influence), **↗** (slight increase), **↘** (slight decrease), **-** (no data available). Meaning of superscript numbers: **1** (for potato starch–lauric acid-complexes).

	Starch														
*Modification*	Properties														
	Structure		Crystallinity		Film forming properties	Hydration properties					Thermal properties			Pasting properties	
	Damaging	Amylose content	Degree	Type	Viscosity	Solubility	Swelling power	Water absorption capacity		Moisture content	Thermal stability	Gelatinization temperatures	Melting enthalpy	Pasting temp.	Final viscosity
*Substitution*	↑	-	loss	loss	-	-	-	-		-	↓	-	-	-	-
*Branching*	↑	↓	↓	B → C	↑	↑	-	-		-	-	↓	↓	-	-
*Debranching*	↑	-	-	→	→	→	-	-		-	→	→	-	-	-
*Hydrolysis*	↗/↑	↓	↑	-	↓	↑	-	-		→	↓	↓/→	→	→	↓
	**Protein**														
*Modification*	Properties														
	Enzyme	Structure				Crystallinity	Rheological properties	Hydration properties		Thermal properties	Foaming properties		Emulsifying properties		
		α-helix content		β-sheet content		Degree		Solubility		Stability	Ability/Capability	Stability	Activity	Stability	
*Cross-linking*	Transglucosidase	→		↑		↓	↑	-		↑	-	-	-	-	
	Peroxidase	↘		↑		↓	↑	-		↑	-	-	-	-	
	Tyrosinase	↘		↑		↓	↑	-		→/↗	-	-	↑	↑	
	Laccase	↘		↑		↓	↑	-		→	-	-	-	-	
*Hydrolysis*	Alcalase	↑		↓		-	-	↑		-	↑	↓	↑	↑	
	Protease	-		-		-	-	-		-	-	-	↗	-	
	**Potato-based films**														
*Modification*	Properties														
	Crystallinity	Film forming properties				Hydration properties					Thermal properties	Tensile properties			
	Degree	Viscosity	G’	Smoothness		Solubility	Moisture content	Water absorption	WVP	Surface hydrophobicity	Tg/Thermal stability	Elongation modulus	Tensile strength	Elongation	
*Debranching*	-	-	-	↑^1^		-	-	-	↓ ^1^	-	-	-	↑ ^1^	↓ ^1^	

## Data Availability

Data is contained within the article.

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
