# Peer review of "Physical, Chemical and Biochemical Modification Approaches of Potato (Peel) Constituents for Bio-Based Food Packaging Concepts: A Review"

_foods, 2022, doi:10.3390/foods11182927_

Round 1
Reviewer 1 Report
It is evident that authors have put a lot of effort into writting this manuscript. However, it resembles more a chapter in the book or the literature riview for thesis than an article. It is too long and too extensive. In the review article, only the new findings should be reviewed, not general and already well established facts. For example, there is no need to describe the modification processes and their influence on starch properties, these have already been reviewed on many occasions. Authors should focus on aspects important for their subject - application in bio-based packaging and give an overview of the literature regarding the use of potato by-products in the production of bio-based packaging in 20 - 25 pages, no more than that.
Author Response
Thank you very much for your feedback. You can find our answers in the attachment.

Reviewer 2 Report
The manuscript " Physical, Chemical, and Biochemical Modification Approaches of Potato (Peel) - Constituents – An Extensive Summary-type Review " extensively reports about potato starch and protein as well as their physical, chemical, and biochemical modification.
The topic of the paper that the authors decided to cover is extremely interesting, but also very broad. The paper was written very carefully with a lot of researched and presented papers, which is commendable, but with too much detail related to the results obtained. Although I understand the need of the author to describe in detail the research conducted and the results obtained, I believe that the manuscript with so much detail is not suitable for reading. It is too extensive and I think 60 pages by far exceeds the usual number of pages of a review work by more than double. I consider such a detailed presentation of the results superfluous and unnecessary. I believe that the readers who will be interested in a particular work from this review will look for it and will read it in its entirety. In my opinion, it is sufficient to present what was studied and to show general conclusions of the paper, without such a large amount of numerical data, which are given even in cases when the results obtained are insignificant.
Detailed comments:
Suggestion is to a little bit alter title in: Physical, Chemical, and Biochemical Modification Approaches of Potato Starch and Protein – An Extensive Summary-type Review
Accordingly, to do the same in whole text
32-33 lines - “56.4 million 31 tons in 2019. Especially in Europe, potato cultivation and potato industry have a high priority in domestic food culture and nutrition. In China the annual potato harvest was 91.818 thousand tons in 2019”
Suggestion is to write numbers on uniform way, both number in thousands or both number in millions
364 line Maybe in one sentence connect the relationship between swelling, pasting I viscosity and HMT conditions
382 – 412lines - extremely detailed description of that research (78) – my suggestion is to point out just the most important data
394-395 lines - After HMT, the umbilical point and surrounding area became darker and with increasing holding time and iterations, granular edges and growth rings became more blurry,… - it seems too detailed to me
460 – “Granule morphology nor the crystalline structure were significant modified through ANN treatment at 30 to 50 °C. Do authors want to say “Granule morphology and the crystalline structure were not significant modified through ANN treatment at 30 to 50 °C.” – If yes, I prefer the second version.
The following are just some examples of too extensive research descriptions but there are many more in the text.
399-400 “which according to the first decreased from 28.9 % down to 10.2 and 10.6 %, but then increased again up to 13 and 25.2 % with increasing iterations and holding time, respectively.”
425-429lines “Karki et al. [15] investigated the effect of HMT (25 – 18 % moisture content, 110 °C, 3 h) on potato waste starch-based films, plasticized with glycerol or sorbitol. When HMT starch was used for film formation, obtained bioplastic films showed slightly decreased solubility and water vapor permeability, as well as slightly increased tensile strength and elongation. However, these differences were not statistically significant, except for the decrease in solubility of sorbitol plasticized films.” – only statistically significant differences should be mentioned
461-463 “However, the degree of crystallinity and crystalline structure were affected by the subsequent drying method. Other than ethylene dehydrated ANN starches, freeze-dried ANN starches showed a change from B-type to C-type crystallinity and a decreased degree of crystallinity.” - it could be said concise
480 “by lower absorbance ratios at 1047/1022 cm-1 and higher full width at half maximum 467 of the band at cm-1 value.” – not necessary
584 “microscopy(PLM)images” - with space
604 “Flexible method” – Does it mean that it is not limited with moisture content?
697 – MV – should be MW?
724 “reviewed by [106] and [107]” – suggestion is to write that just: [106, 107]
745 “Due to HP treatment (50 - 1000 MPa for 1 h) potato starch granules were found to become more angular and their bulk density clearly increased.” – rephrase this sentence that would be clear what was treated
762-767 “Gelatinization temperatures were slightly lower for starch suspensions treated at lower pressure (30 and 60 MPa), but similar or slightly higher for starch suspensions treated at higher pressure (90 764 and 120 MPa), compared to native potato starch suspensions. Degree of gelatinization increased with increasing pressure and final viscosity increased upon HP microfluidization treatment by max. 500 mPa s. – Is it possible to say that more concise e.g. “Gelatinization temperatures and degree of gelatinization increased with increasing pressure and final viscosity increased upon HP microfluidization treatment by max. 500 mPa s.”
771 “of emulsions stabilized by potato protein isolate” – it seems to me as not necessary to be written
776 – “DSC analysis revealed a gradual increase in denaturation temperature up to 5.6 °C and decrease in ΔH by half with increasing pressure.” – it seems as to much details
779 – “Baier and Knorr [102] studied the influence of high isostatic pressure (200 - 600 MPa) – description of that study is to detailed
810 (> 60 °C).Increasing – space is missing
893 “FT-IR spectroscopy revealed a gradual decrease of IR ratio 1047 cm−1/1022 cm−1 by single frequency treatment and dual-frequency treatment, which further decreased at elevated temperature, 895 indicating induced damages to the double helix structure by US treatment. “- it could be said concise
896-897 “Based on X-ray diffraction patterns, Hu et al. (2019) reported C-type crystallinity of potato starch samples, which differ to the previous reports of typical B-type crystallinity [47]. However, - it seems to me as not necessary to be written
905 nor the combination of both significantly affected the initial protein solubility (approx. 30 %). Did it significantly influence or not? Please rephrase it to be completely clear
937 – 946 lines – right margin
944 – “elongation and decreased” – and?
Author Response

(The authors gave the same response as above.)

Reviewer 3 Report
This is the revision of the manuscript with the title "Physical, Chemical, and Biochemical Modification Approaches of Potato (Peel)-Constituents – An Extensive Summary-type Review" by Miller K. & colleagues. The review gives information on the physical, chemical, and biochemical modification approaches of potato peel constituents. It is a well written and well organised overview of the publications available online, mostly in the last five years. Special attention is devoted to the discussion of the modification techniques, with a focus on the changes of selected properties as a result of applying a given modification method. It is a very useful and comprehensive review. However, I would recommend improving Figure 1. The quality of it is low. Moreover, the abstract is too long. According to editorial requirements, an abstract should not be longer than 200 words. Please rewrite this text to meet the requirements.
Author Response

(The authors gave the same response as above.)

Round 2
Reviewer 1 Report
The authors have somewhat reduced the volume of the article, but it is still too comprehensive for an article. The article is not diveded into starch- and protein-part, which contributes to difficulty of reading since we are going back and forth from starch to protein. Furthermore, a lot of modification procedures are in detail reviewed elsewhere and there is no need to repeat these. The authors should focus on the novel approaches and solutions.
Author Response
Thank you very much for your comment. We are sorry that we were not able to convince you even after the intensive shortening and revision of our manuscript. As explained in our last revision, there is currently no review article dealing with the (different) modification potentials for the potato side stream(s). For this reason, we also chose to structure the review along the modification methods that were summarized and compared later in the review.
Your idea to modify the structure and content of the review is interesting and would also lead to a very interesting review article, but with a focus that, at least from our point of view, no longer corresponds to our actual intention for the article. The current manuscript has been evaluated very positively by other reviewers, also in terms of relevance, so we think that there will also be an interested readership for the article in its current form.
We hope you understand that we have kept the structure of the manuscript and the modification methods, and we thank you again for taking the time to review and provide feedback.
With best regards,
the authors
Reviewer 2 Report
Congratulations to the authors for a great job. It is a remarkable improvement. Manuscript is now much better. Still there are some long parts which could be shorter e.g. 287 – 297, 345 –352, 406 -414, 420 – 424. Also, in line 457 “by Castro et al. [104] and Yang et al. [105]”, the shorter version is just [104, 105]. Generally, sentence in lines 501 – 503 is unnecessary. In tables references are missing. α helix should be written uniformly through the whole text and tables (with or without a dash).
Author Response
Dear Reviewer,
Thank you very much for your valuable feedback and positive evaluation.
We have shortened the paragraphs you mentioned and changed the reference in line 457 "from Castro et al [104] and Yang et al [105]" to [104,105].
We also deleted the sentence in lines 501-503, wrote α-helix consistently throughout the manuscript, and added the appropriate references to Tables 2-4.
With best regards,
the authors